# Lobule-Related Action Potential Shape- and History-Dependent Current Integration in Purkinje Cells of Adult and Developing Mice

**DOI:** 10.3390/cells12040623

**Published:** 2023-02-15

**Authors:** Gerrit C. Beekhof, Martijn Schonewille

**Affiliations:** Department of Neuroscience, Erasmus MC, 3015 AA Rotterdam, The Netherlands

**Keywords:** purkinje cell, action potential, current integration, aldoc, zebrinii, lobule specific, hcn, cerebellum

## Abstract

Purkinje cells (PCs) are the principal cells of the cerebellar cortex and form a central element in the modular organization of the cerebellum. Differentiation of PCs based on gene expression profiles revealed two subpopulations with distinct connectivity, action potential firing and learning-induced activity changes. However, which basal cell physiological features underlie the differences between these subpopulations and to what extent they integrate input differentially remains largely unclear. Here, we investigate the cellular electrophysiological properties of PC subpopulation in adult and juvenile mice. We found that multiple fundamental cell physiological properties, including membrane resistance and various aspects of the action potential shape, differ between PCs from anterior and nodular lobules. Moreover, the two PC subpopulations also differed in the integration of negative and positive current steps as well as in size of the hyperpolarization-activated current. A comparative analysis in juvenile mice confirmed that most of these lobule-specific differences are already present at pre-weaning ages. Finally, we found that current integration in PCs is input history-dependent for both positive and negative currents, but this is not a distinctive feature between anterior and nodular PCs. Our results support the concept of a fundamental differentiation of PCs subpopulations in terms of cell physiological properties and current integration, yet reveals that history-dependent input processing is consistent across PC subtypes.

## 1. Introduction

The cerebellum receives and integrates a wide range of information, which it uses to control and coordinate movements as well as various cognitive and emotional processes. This enormous integrative power is visualized by the massive dendritic tree of the cerebellar Purkinje cell (PC), estimated to receive over 100,000 inputs. On top of that, PCs are intrinsically active and continuously send their inhibitory output, the only output of the cerebellar cortex, to the cerebellar nuclei [1]. The principles of integrating inputs, excitatory and inhibitory, with the intrinsic activity, are critical for understanding cerebellar functioning. The cerebellar organization is based on modules, circuits consisting of connected neurons in the inferior olive, cerebellar cortex, and cerebellar nuclei (CN) [2,3,4,5]. Distinct areas control specific tasks, such as compensatory eye movements [6,7,8,9], controlling limb and finger movements [10,11], and associative motor learning [12,13,14,15,16]. Cerebellar modules can be distinguished by different levels of molecular markers in PCs [17,18], of which Aldolase C (Aldoc) (also known as ZebrinII (Z)) is widely used [19,20], and is evolutionarily well conserved across species [21,22,23,24]. These PC modules are not only histologically but also morphologically [25,26] and physiologically distinct [27,28,29], already early in development [26]. PCs project to discrete targets in the vestibular and cerebellar nuclei [30,31,32], and cause Aldoc related firing in the CN [33]. PC modules are linked to specific muscle groups [34,35], receive climbing fiber input from specific inferior olive subnuclei [36,37,38], show zone dependent synaptic plasticity [39,40], and are linked to neurodegenerative diseases [41,42]. Cerebellar learning in PCs results in an increased [43,44,45] or decreased [46,47] firing rate activity in relation to adaptation. PCs in Aldoc-negative (Aldoc− or Z−) cerebellar modules, which have intrinsically a higher firing rate compared to Aldoc-positive (Aldoc+ or Z+) PCs, show a decrease in firing rate during the learned response, while an increase in firing rate upon learning is observed in PCs of Aldoc+ modules [48]. The ability of PCs to integrate inputs and thereby spatiotemporally modulate their firing rate and motor behavior has been explored extensively in the adult, but only sparsely during development of the circuit formation [49,50,51,52]. Given the robust differentiation of the two populations, it is crucial to understand the intrinsic and integration properties of the individual PCs between zones in both adulthood and developmental stages.

The tonic firing rate in PCs is mainly generated by the inward current of voltage-activated sodium channels [53]. Tonic intrinsic firing rates differ between PCs depending on their zonal location [27,28]. The PC zones are biochemically distinct and differential expression of channels such as the transient receptor potential cation channel C3 (TRPC3) [54] and the small conductance Ca^2+^-activated K^+^ channels type 2 (SK2) [55,56] contribute to differences in tonic intrinsic activity. During postnatal maturation PCs increase the capacity to fire action potentials (APs) by lowering their action potential threshold [57]. As the PC subpopulations differ in the expression of a large number of genes [17,18,29], we hypothesize that other intrinsic properties also differ between populations of (PCs). One of those properties is the history-dependent integration of inputs, a mechanism that was previously found in PCs [58]. History dependent integration indicates a sensitivity to the previous state of the membrane potential and timing of the applied input and this can be considered as a cell-based short-term memory. It has been suggested that the learning-related cellular activity of Aldoc− PCs consists of a decrease in simple spike firing, while that of Aldoc+ (or phospholipase C β4− (PLCb4−)) PC is marked by an increase in firing rate [48]. This zonal specific learning is supported by the findings in juvenile mice that Aldoc− PCs have a higher intrinsic excitability, and an enhanced intrinsic plasticity and synaptic long-term potentiation compared to Aldoc+ PCs [59] and that the Aldoc− PC have a higher level of plasticity in terms of gene expression [29]. This and earlier work indeed revealed higher expression levels of various ion channels and proteins related to second messenger pathways, and calcium storage/release [60]. Not surprisingly, history-dependent input–output dynamics are controlled by these types of intrinsic neuronal properties [61,62], raising the question whether these important PC features also vary between subpopulations. If history-dependent dynamics indeed differ between subpopulations of PCs, this could underlie the distinct contributions of PCs to learning related to different cerebellar microzones.

Here, we test the hypothesis that the basic physiological properties, including history-dependent integration, are different in subpopulations of PCs and determine when these differences are established. We found that several features of the action potential shape differ between PCs from anterior lobule III (LIII) and nodular lobule X (LX), and between PCs from postnatal day 30 (P30, mostly adult) and older and those from juvenile mice (P3–5, P15–17 and P22–24). Next, we show that in >P30, but not juvenile mice, anterior PCs can be driven to higher levels of depolarization and higher firing frequency, while LX PCs can be more hyperpolarized. Finally, we found that current integration in PCs is input history-dependent for both positive and negative currents, but that this is not a distinctive feature between anterior and nodular PCs. Overall, this study shows that PC subpopulations are unique in intrinsic electrophysiological properties, and that this is dependent on postnatal age.

## 2. Materials and Methods

### 2.1. Mice

All animals in this study were handled and kept under conditions that respected the guidelines of the Dutch Ethical Committee for animal experiments and were in accordance with the Institutional Animal Care and Use Committee of Erasmus MC (IACUC Erasmus MC), the European and the Dutch National Legislation. All animals were maintained under standard, temperature controlled, laboratory conditions. Mice were kept on a 12:12 light/dark cycle and received water and food ad libitum. We recorded PCs in male and female C57BL/6J or *Slc1a6-EGFP* (*Tg(Slc1a6-EGFP)HD185Gsat/Mmucd*) transgenic mice [63,64].

### 2.2. In Vitro Recording and Analysis

The in vitro extracellular recordings were performed in either *Slc1a6-EGFP* [64,65] or C57BL/6J mice, ranging in age from P3 to P240. All PCs in lobule X (LX) are Aldoc+ and virtually all PCs in lobule III (LIII) are Aldoc−. In ~60% of the experiments we used *Slc1a6-EGFP* mice to verify that the recorded LIII PCs are indeed Aldoc−, minimizing the potential risk of including Aldoc+ PCs in the LIII dataset. PCs were identified based on their large soma size and presence in the PC layer. As previously described [26], the brain was quickly removed and placed in ice-cold slice solution (continuously carbogenated with 95% O_2_ and 5% CO_2_) containing the following (in mM): 240 Sucrose, 2.5 KCl, 1.25 NaH_2_PO_4_, 2 MgSO_4_, 1 CaCl_2_, 26 NaHCO_3_, and 10 D-glucose. Acute sagittal slices of vermal cerebellar tissue of 250 µm thick were cut in ice-cold slicing solution using a vibratome (VT1000S, Leica Biosystems, Wetzlar, Germany) with a ceramic blade (Campden Instruments Ltd., Manchester, United Kingdom). Directly after slicing, the slices were transferred to a recovery bath and were incubated in oxygenated artificial cerebrospinal fluid (aCSF) and maintained at 34 °C for one hour. The aCSF was continuously carbogenated with 95% O_2_ and 5% CO_2_ and consisted of (in mM): 124 NaCl, 5 KCl, 1.25 Na_2_HPO_4_, 2 MgSO_4_, 2 CaCl_2_, 26 NaHCO_3_, and 20 D-glucose. After the incubation period, slices were transferred to room temperature. To record the individual slices, these were transferred to a recording chamber and maintained at 34 ± 1 °C with a feedback temperature controller with heater (Scientifica, Uckfield, UK) under continuous superfusion with the oxygenated aCSF.

PCs were visualized with SliceScope Pro 3000, a CCD camera, a trinocular eyepiece (Scientifica, Uckfield, UK) and ocular (Teledyne Qimaging, Surrey, BC, Canada). Whole-cell and cell-attached recordings were obtained using borosilicate pipettes (Harvard apparatus, Holliston, MA, USA) with a resistance of 4–6 MΩ, filled with internal solution containing (in mM): 9 KCl, 3.48 MgCl_2_, 4 NaCl, 120 K+-Gluconate, 10 HEPES, 28.5 Sucrose, 4 Na2ATP, 0.4 Na3GTP in total pH 7.25–7.35, and osmolarity 290–300 mOsmol/Kg (Sigma-Aldrich, Merck KGaA, Darmstadt, Germany).

Recording pipettes were supplemented with 1 mg/mL biocytin to allow histological staining, for morphological analysis of PCs over different developmental stages see Beekhof et al. (2021) [26]. Whole-cell and cell-attached recordings were performed using an ECP-10 amplifier (HEKA Electronics, Lambrecht, Germany) and digitized at 50 kHz. Acquisition was carried out in Patchmaster (HEKA Electronics, Lambrecht, Germany) and ABF Utility (Synaptosoft, Fort Lee, NJ, USA) was used to convert the Patchmaster files for analysis. Clampfit 10 (Molecular Devices, LLC, San Jose, CA, USA) was used to analyze AP firing rates and a custom-build MATLAB code (Matlab 2016, MathWorks, Natick, MA, USA) using inter spike properties to analyze AP variables [66] and AP shape properties. The coefficient of variation 2 (CV2) represents the variance on a spike-to-spike base, calculated by dividing the absolute difference between two consecutive inter-spike-intervals (ISIs n and n+1) by their mean (giving the CV per interval) and averaging this value over all intervals; 2*|ISI_n+1_ − ISI_n_|/(ISI_n+1_ + ISI_n_).

For both whole-cell and cell-attached recordings, slices were bathed in aCSF supplemented with synaptic receptor blockers, NMDA receptor antagonist D-AP5 (50 µM), selective and competitive AMPA receptor antagonist NBQX (10 µM), non-competitive GABA_A_ receptor antagonist and glycine receptor inhibitor picrotoxin (100 µM) (all chemicals from; Hello Bio Ltd., Bristol, UK).

### 2.3. Measurement of In Vitro Electrophysiological Parameters

To record the AP waveform, we performed whole cell recordings, we created a gigaseal between the electrode and the PC membrane. After breaking the gigaseal the input/series resistance were determined by performing a seal test protocol, a voltage step of −10 mV averaged out of 7 sweeps, followed by an analysis of the rheobase. The rheobase is determined by holding the cell in current clamp at −65 mV and each second increasing the current with 1 pA until the cell fires its first action potential. This first AP is analyzed to calculate action potential properties. The first AP is isolated and analyzed, the AP threshold was calculated as the potential at which the second derivative peaked (d^2^V/dt^2^) [67]. The baseline for analyzing the amplitude of the AP and the fast afterhyperpolarization (fAHP) were calculated relative to the mode of 10 msec of the trace (from −15 to −5 msec before the AP peak). The half-width, rise, and decay time were calculated by using the AP threshold at the start of the AP, using 10–90% of the total rise and decay period to calculate the time. Note that the values of these parameters approach the sampling rate (50 kHz), resulting in a step-wise pattern of values. The AP amplitude is calculated from the baseline to the AP peak, and the AP maximum is the membrane voltage at the peak of the AP. The fAHP time is the time it takes after the AP from the baseline to the lowest point of the fAHP, when there is no fAHP we reported an amplitude of 0 mV with no time point. If the trace during the rheobase protocol was unstable or showed a spikelet before the AP we did not include this AP in the analysis. For some, but not all cells, we proceeded recordings with current steps and/or ramps, and not for all cells for which we report a current step and/or ramp experiment, the rheobase was included in this dataset.

To record the current integration for negative and positive steps, we performed whole cell recordings in vitro and executed a negative and positive current step protocol in PCs. For the negative current step protocol, we clamped the cell at −55 mV which allows the cell to fire action potentials, and switched to current clamp mode and injected negative current in steps of −40 pA from 0 pA to −760 pA. For the positive current step protocol, we clamped the cell at −65 mV at which the cell does not fire, and switched to current clamp mode and injected positive currents in steps of 40 pA to from 0 to 760 pA. Both the negative and positive current injection lasted a second, and during this time we measured the mode of the membrane potential and the firing frequency. Before and after the current step we injected 2 s of 0 pA to give the cell time for recovery of the current steps.

To record the history dependent integration of currents, we performed whole cell in vitro recordings and executed a negative and positive current ramp protocol. The negative current ramp protocol required us to clamp the cell at −55 mV, we then switched to current clamp mode and injected a negative current ramp from 0 pA to −320 pA to 0 pA. Both the downphase from 0 pA to −320 pA and the upphase from −320 pA to 0 pA took 1.5 s, therefore the total current ramp injection takes 3 s to complete. For the positive current ramp protocol, we clamped the cell at −65 mV, then we switched to current clamp mode and injected a positive current ramp from 0 pA to 320 pA to 0 pA. Here also both the upphase from 0 pA to 320 pA and the downphase from 320 pA to 0 pA took 1.5 s each. Before and after the injection of current we injected 2 s of 0 pA to give the cell time for recovery of the current ramps. The ramping currents were repeated 10 times and averaged, for the positive current clamp we included an extra quality control, the PCs must stop firing after the injected current to be sure that the PCs did not escape the clamp, only recordings with at least 6 traces of proper quality were included in our analysis. For the negative and the positive current ramp protocol we binned the trace in bins of 0.2 s, so that the biggest current, −320 pA or 320 pA was in the middle of a bin. We calculated the mode of the membrane potential and the average firing rate for each bin. Note that the reported currents for the current step and the current ramp protocol are relative to the current needed to hold the cell at −55 mV of −65 mV.

### 2.4. Measurement of In Vitro Electrophysiological Parameters—HCN Currents

To measure the hyperpolarization-activated cyclic-nucleotide-gated channel (HCN) currents, the following chemicals were added to the bath on top of the above-mentioned synaptic receptor blockers; voltage-gated Na^+^-blocker TTX (1 µM) (Hello Bio Ltd., Bristol, United Kingdom), Kir blocker BaCl_2_ (1 mM), voltage-gated Ca^+^-blockers NiCl_2_ (1 mM), and CdCl_2_ (0.1 mM). For measuring the HCN currents after blocking HCN1, propofol (100 µM, European Pharmacopoeia Reference Standard) a HCN1-blocker [68] was released in the recording bath. Blocking the total amount of HCN currents, was performed by fusing the general HCN-blocker zatebradine hydrochloride [69] (50 µM) in the recording chamber (all above chemicals from; Sigma-Aldrich, Merck KGaA, Darmstadt, Germany).

HCN currents were measured in voltage clamp, the voltage was clamped at −45 mV for 2.5 s before and after the voltage step. The voltage steps did have a duration of 500 msec and decreased in 17 sweeps from −45 mV to −130 mV, at each following step the voltage step was increased by −5 mV. The total amount of current to make the voltage step possible was measured, as also the tail current. The total amount of current to facilitate the voltage step is not representative for the number of channels opening or closing, therefore we report these values only in the Appendix A. The tail current is a current which is still visible after terminating the voltage step. Closure of a channel is not abrupt which allows ions to pass the channels until they are closed, this current measured is the tail current and represents the number of active channels upon a voltage step. Here the tail current is representing HCN currents, since all other channels are blocked. The mode of the current at −45 mV, from 1 s to the voltage step, was taken as the baseline. The mode of the last 50 msec of the voltage step was taken as the baseline during the voltage step, and the difference between the baseline before and during the step was taken as the step current measurement. The tail current is measured as the difference between the baseline before the voltage step, as reported before, and the trough of the tail current. Since the closure of the HCN channels is relatively slow [70], we took the 1600 lowest values at the trough and calculated the median of these 1600 values. This step assures that eventually incoming noise is deselected, and provides an accurate report of the actual tail currents.

Cell attached recordings were made with a seal of 30 MΩ to 2 GΩ and lasted for a minimum of 90 s up to 150 s. For recordings measuring intrinsic PC activity without HCN1 current we added HCN1 blocker propofol (100 µM, European Pharmacopoeia Reference Standard, Sigma-Aldrich, Merck KGaA, Darmstadt, Germany) and selective GABA_B_ receptor antagonist CGP52432 (50 µM, Sigma-Aldrich) to block the agonistic effects of propofol on GABA_B_ receptors [71]. A control experiment to rule out the effect of CGP52432 (CGP) was also performed. Whole cell recordings, to measure the AP shape were made as described above, now only with the addition of propofol (100 µM) added to the bath. We did not add CGP, since we did not expect significant effects on the AP shape based on the previous cell attached recordings. Mice used for HCN related cell attached recordings were not used for HCN related AP shape recordings, and vice versa.

### 2.5. Statistical Analysis

Error bars in all graphs indicate the SEM. Data were tested for normality with the Shapiro–Wilk test and for equal variances we used the F-test. When data were normally distributed, we used the two-way ANOVA with multiple comparisons or mixed-effects test with repeated measures to determine statistical significance. When data were not normally distributed, we used the Kruskal–Wallis test with multiple comparisons to determine statistical significance. For all tests we used a minimal level of significance of *p* < 0.05. We used GraphPad Prism 9.5 (GraphPad Software, San Diego, CA, USA) to execute statistical analyses.

## 3. Results

To compare the cell intrinsic features between two distinct PC subpopulations, we recorded PCs from LIII and LX, proxies for Aldoc− and Aldoc+ PCs, respectively. All PCs in LX are Aldoc+ and nearly all PCs in LIII are Aldoc−. We performed whole cell recordings in vitro in PCs of C57BL/6J mice or, in a subset of experiments, of *Slc1a6-EGFP* mice (Figure 1(A1)). *Slc1a6-EGFP* mice express the enhanced green fluorescent protein (EGFP) under the *Slc1a6* promotor, an expression pattern that correlates with high levels of Aldoc (or ZebrinII) [63,64]. PCs were identified based on large soma size and their localization in the PC layer, and in *slc1a6-EGFP* mice the identity was online verified.

### 3.1. Purkinje Cell Membrane Resistance and Action Potential Shape Differ between Subpopulations

Following previous work that found differentiation in action potential firing and morphological features of PCs [26,27], we first looked for potential differences in basic properties and action potential shape. We initially made recording from slices of mice older than 30 days (postnatal day 30, >P30), as our previous work demonstrated that the electrophysiological development is completed by this age [26]. After breaking the gigaseal, input and series resistance were determined by performing a seal test protocol (Figure 1(A2)), followed by an analysis of the rheobase (Figure 1(A3)). The first AP of the rheobase test was analyzed to calculate action potential properties (Figure 1(A4)) for both groups. The first observation was that the input resistance is significantly lower in LIII PCs compared with the LX PCs (*p* < 0.0001, Figure 1B). This difference is not an artifact as it is based on a difference in membrane, not series resistance (Figure 1 and Appendix A). This is in line with the finding that PCs in LX are on average smaller than those in LIII PCs, as LX PCs are shorter, have a lower membrane area surface, and are less complex than LIII PCs [26] and smaller cells tend to have a higher input resistance. Moreover, LIII PCs require a larger current injection to elicit the first AP (*p* = 0.0304, Figure 1C). The difference in rheobase is not caused by a difference in the AP threshold (*p* = 0.3809, Figure 1D), which could indicate that the differences in input resistance and rheobase are caused by, e.g., the difference in size of the PC or the Na^+^-channel density. Next, we looked in detail at the shape of the AP (Figure 1(A4)) in both populations. We observed a larger AP amplitude in LX PCs (*p* = 0.0002, Figure 1E), and AP peak (*p* < 0.0001, Figure 1 and Appendix A). The AP waveforms of LIII and LX PCs have a similar rise time (*p* = 0.1621, Figure 1(F2)), but the half width and decay time are longer in PCs in LX (*p* = 0.0011 and *p* < 0.0001, respectively, Figure 1(F1,1F3)). The amplitude of the fast afterhyperpolarization (fAHP) in PCs is larger in LIII with a shorter time to peak in comparison to LX (*p* = 0.0346 and *p* = 0.0005, Figure 1G and Appendix A).

Taken together, our results show that, on top of differences in firing rate, there are more fundamental differences in the PCs in membrane resistance, rheobase, AP amplitude, half width, and after hyperpolarization. Together these differences suggest that the gene expression differences are not only responsible for distinct activity levels, but also affect excitability and core features of action potential shape.

### 3.2. Current Injection-Evoked Bidirectional Changes in Firing Rate and Membrane Potential Differ between PC Subpopulations

PCs are intrinsically active, even in the absence of their inhibitory and excitatory synaptic inputs [53,72,73], and PCs in Aldoc− microzones have a higher firing frequency than those in Aldoc+ microzones [26,27]. Next, we compared the effect of current injections, positive and negative, on PC firing frequency and membrane potential between lobules. To answer this question, we subjected PCs during whole cell recordings in slice to a negative (Figure 2(A1)) and positive (Figure 2(A2)) current step protocol (see Methods). When PCs were held at −55 mV, the firing frequency between LIII and LX was significantly different. Upon initiation of the negative current step protocol, both in LIII and LX the AP firing stops in virtually all PCs at the same moment when −120 pA is injected (Figure 2(B2)). However, normalizing the firing rate to the rate at the start (0 pA), the firing rate drops faster in LX than in LIII PCs (Figure 2(B3)), in line with the higher input resistance (Figure 1B). When we examined the underlying, membrane potential, we found that it starts to differ between LIII and LX from a negative current injection of 280 pA onwards (Figure 2(B1)), with PCs in LX showing stronger hyperpolarization. Upon injection of positive current injections, PCs in LIII depolarize from 640 pA onwards (Figure 2(C1)). Normalizing the firing rates to the rate at maximum current injection (+760 pA), again confirmed the faster change in rate in LX (Figure 2(C3)). Nonetheless, we find that in terms of absolute firing rates, PCs in LIII are significantly more excitable than PCs in LX starting from current injections of 320 pA (Figure 2(C2)).

Next, we determined the voltage sag in response to hyperpolarizing current step injections, which is indicative for the presence of the hyperpolarization-activated (Ih) current [74]. Ih is needed to maintain the membrane potential in a range needed to sustain tonic firing of PCs [58], suggesting that it could be involved in controlling firing rate and excitability. Indeed, we found a difference in the voltage sag suggesting a different level of maintaining tonic firing in Aldoc PC populations (Figure 2 and Appendix A). Contrary, we found no differences in the calcium-activated slow afterhyperpolarization (sAHP), occurring in response to depolarizing current step injections [75] (Figure 2 and Appendix A).

Taken together, our results indicate that, even though positive current injections do not result in differences in the membrane potential, firing rates are higher in LIII PCs, suggesting that membrane potential does not directly determine the firing frequency in the PCs of LIII and LX. We also show that LX PCs are prone to have larger changes in membrane potentials compared to LIII PCs upon the same negative current injection steps, which is in line with their higher input resistance.

### 3.3. Purkinje Cells of Both LIII and LX Show History Dependent Integration of Current Inputs

After finding that the PC firing rate and action potential shape, as well as the membrane potential of PCs are different in PCs from LIII and LX, we asked if the state of the membrane potential and the firing rate is dependent on the previous amount of injected current for PCs in LIII and LX. Previous work on history dependency integration of inputs in PCs yielded contrasting results; one study shows that the firing rate is independent of the injected current [58], while another shows that the membrane potential and firing rate are dependent on the injected current [76]. However, the studies used mice from different developmental ages and did not take the Aldoc-identity into account, while we have recently shown that these underlie significant variations between PCs [26].

To answer this question, we performed whole cell recordings in vitro and injected negative and positive ramp currents in PCs (Figure 3A). Unsurprisingly, the negative current ramp shows that the membrane potential of PCs in LIII is less hyperpolarized compared to LX with the same amount of current injected (−320 pA: *p* < 0.0001, Figure 3(B1)), while the action potential firing stops virtually at the same time (Figure 3(C1)). When we plot the hyperpolarizing phase (downphase) and the depolarizing phase (upphase) on top of each other, we find that the two curves do not overlap, indicating that the amount of injected current does not directly cohere with the membrane potential in PCs, both in LIII and LX (Figure 3(B2,B3)). This history-dependence in membrane potential is also reflected in the PC firing frequency in both LIII and LX (Figure 3(C3)). Here too, the range of current injections for the negative current ramp during which there is a significant difference in membrane potential between the downphase and the upphase, does not exactly match that for firing rate, indicating, again, that the membrane potential is not directly coupled to the firing frequency.

Upon injecting positive ramping current, we found a subtle but significant difference in membrane potential between LIII and LX PCs at low current injections (Figure 3(D1)). Complementary to the positive current step, we found a significant firing rate difference between LIII and LX at 320 pA (320 pA: *p* < 0.0001, Figure 3(E1)). Plotting the depolarizing phase (upphase) and hyperpolarizing phase (downphase) on top of each other, we find that, here too, the membrane potential is dependent on the previous state of the membrane potential and current injection in both LIII and LX (Figure 3(D2,D3)). The same concept applies to the firing rate of the PCs in LIII and LX (Figure 3(E2,E3)). Two features stand out here: (1) there is a consistent delay in both membrane potential and firing rate, relative to the change in current, and (2) the differences in downphase versus upphase extend across a larger part in the negative current ramp than in the positive ramp.

Taken together, PCs show history-dependent current integration, suggesting that the previous state of the cells is relevant for the processing of new inputs. While the intrinsic firing activity is robustly different between the two subpopulations, the differences in history-dependent current integration are present, but more subtle.

### 3.4. Development of Basic Purkinje Cell Features and Action Potential Shape

The cerebellum, and cerebellar PCs in particular, develop and reach mature status relatively late [77,78]. We and others have shown that PC firing rate continues to increase until at least week 4 [79] and can take even longer in Aldoc− PCs [26,77]. As basic PC features and action potential shape are also different between PC subpopulations, next we investigated when the differences between LIII and LX in basic PC features and action potential shape appear during development. To cover postnatal development, we recorded PCs in three different age groups: an early postnatal group of postnatal days 3 to 5 (P3 to P5), a late postnatal group aged P15 to P17, and a juvenile group aged P22 to P24. We were only able to obtain a relatively small number of PCs for LIII at P3–P5. Therefore, we will not directly compare LIII to LX PCs and restrict our conclusions to comparing P3–P5 PCs to older ages. The morphology of PCs changes drastically during maturation (Figure 4A). PCs of P3–P5 mice are morphologically under-developed, these neurons are very small compared to fully developed PCs. Small neurons typically have a higher input and membrane resistance, which is indeed confirmed when we compared LIII and LX P3–P5 PCs to LIII and LX PCs of P15–P17 and P22–P24 PCs (Figure 4 and Appendix A). Interestingly, the differences in membrane and series resistance observed in >P30 PCs, was not present at any of the developmental ages (Figure 4 and Appendix A). In line with the >P30 group, there is no difference in AP threshold between LIII and LX in any of the age groups. However, from P3–P5 to P15–17, the AP threshold decreases in both LIII and LX PCs (Figure 4B). The difference in AP amplitude between LIII and LX is not yet developed at any of the ages, but the level changes from P3–P5 to older ages (Figure 4C and Appendix A for AP peak). The AP shape of PCs at P3–P5 in both lobules is dramatically different from older ages in that the half width, rise time, and decay time are significantly longer than those in P15–P17 and P22–P24 mice (Figure 4(D1–D3)). Here, the differences observed between LIII and LX PCs in >P30 mice, can only partially be retrieved at earlier ages. For instance, the longer decay time in LX compared to LIII in >P30 PCs is already established at P15–P17, while the half width of the AP does not reach significance at any developmental age, despite a similar trend in P22-24 PCs as seen in >P30 mice (Figure 4(D1–D3)). Finally, unlike PCs in >P30 mice, the amplitude of the fAHP is not different between LIII and LX PCs at any age, although it should be noted that, in general, AHPs do not yet occur in P3–P5 PCs (Figure 4E). However, the faster kinetics of the fAHP found in LIII PCs >P30 compared to LX are also found in developing mice at P22–P24 but not in PCs of P15–P17 (Figure 4 and Appendix A). The striking absence of an fAHP in P3–P5 PCs may be explained by the fact that these PCs are significantly less mature in terms of morphological and electrophysiological features [26,80,81].

These results indicate that apart from the decay time, most differential features do not appear until after P30. The P3–P5 PC AP waveform is clearly different from PCs at all other ages, presumably due to their small soma and undeveloped dendritic tree.

### 3.5. Response to Current Injections in Developing Purkinje Cells of Distinct Cerebellar Regions

Since many fundamental properties of a PC are not fully matured until after P30, we asked if the same principle applied to the integration of negative and positive step currents. We opted to analyze PCs in juvenile mice at the age of postnatal day 15–17 (P15–P17), because PCs have matured enough to have a functional dendritic tree, yet are significantly younger than the >P30 age group. All recorded PCs were subjected to the same experimental protocol as described before for the >P30 age group regarding the positive and negative current injection steps. The application of negative current injection steps, starting from −55 mV, revealed that the membrane potential of LX is more rapidly hyperpolarized compared to LIII PCs. The difference is significant from −320 pA onwards (Figure 5(A1)). Interestingly, unlike in >P30 PCs, the firing rate of LIII and LX PCs is similar at −55 mV, while LIII PCs stop firing with less injected current compared to LX PCs (Figure 5(A2)). Upon positive current injection steps, the PCs in LIII of >P30 mice are more excitable in that they exhibit a higher firing rate compared to LX PCs (Figure 2(C2)). This difference is absent in PCs of mice that are P15–P17 group (Figure 5(B2)), which is surprising given the differences in resting rate found using cell-attached recordings [26], but may be in line with the absence of a difference in membrane resistance (Figure 4 and Appendix A). Comparing the Ih-related voltage sag and the sAHP, we find contrasting results in P15–17 compared to >P30 PCs. Unlike in >P30 PCs, there is no evidence for a difference in the voltage sag between lobules, suggesting Ih is comparable during development (Figure 5 and Appendix A). In contrast, the amplitude of the sAHP is significantly different at several injection amplitudes between lobules, with a larger sAHP in LIII P15–17 PCs (Figure 5 and Appendix A), while a subpopulation difference was not found in >P30 PCs. Interestingly, both the levels of the voltage sag and the sAHP are 1.5–2 times higher in P15–P17 PCs compared to >P30, suggesting a strong developmental regulation of both.

Taken together, our results indicate that at P15–P17, the differential responses, in particular to positive current injections, are less apparent. In contrast, the levels of the voltage sag and sAHP are substantially higher and show inverted differences compared to >P30 PCs, with differences in sAHP and comparable levels of voltage sag at P15–P17

### 3.6. Developing Purkinje Cells Show History Dependent Current Integration

Given that we demonstrated that P15–17 PCs react differently upon step current injection compared to >P30 PCs, we wondered if history-dependent current integration is also different. We used the same experimental protocol as described before for the >P30 age group, consisting of a positive and negative ramp current injection. Similar to the >P30 group we observed at P15–P17 that LX PCs are more hyperpolarized with the same amount of current compared to LIII PCs (Figure 6(A1)), but at P15–P17 the firing rate between LIII and LX PCs is not different (Figure 6(B1)). For both LIII and LX PCs, we observed the same history dependent integration of a negative ramp current input as in the >P30 PCs, when comparing the hyperpolarizing (downphase) and depolarizing (upphase) trajectories (Figure 6(A1,A2,B1,B2)). For the positive current injection, we found that LIII PCs stop firing APs earlier compared to LX with the same amount of current injected (Figure 6(C1)), as was seen in >P30 PCs. Here too, the results from P15–P17 PCs are comparable to those observed in >P30 PCs (Figure 6C,D). The only clearly contrasting result is that the membrane potential in LIII is not exhibiting the history dependency in P15–P17 LIII PCs (Figure 6(C2)), whereas the firing rate across that section does show similar differences as seen in >P30 PCs. These results suggest that in P15–P17 PCs, at relatively small ramping current injections, the membrane potential is not directly coupled to the firing rate of PCs.

In conclusion, history-dependent current integration is largely similar between P15–17 and >P30 PCs, except for the membrane potential upon small positive ramp current injections, which lacks history dependent integration.

### 3.7. Differential HCN Currents in Purkinje Cell Subpopulations

Previous research shows that all hyperpolarization-activated cyclic-nucleotide-gated channel (HCN) subtypes, HCN1-4, are expressed in PCs [82] and that HCNs are involved in the bistability of PCs [58]. It has been demonstrated that, although HCN1 does not directly affect AP shape and firing rate [76], it does control various aspects of neural integration, including the integration of inhibitory inputs [83]. We previously found that adult PCs of LIII not only have a higher firing rate, but also a higher coefficient of variation (CV) compared to LX PCs [26]. Therefore, we wondered if a differential functional contribution of HCNs in general and HCN1 specifically in PCs could underlie the lower CV2 in LX PCs or any of the current integration differences observed in the current study. To answer this question, we performed whole cell recordings in vitro in voltage clamp in >P30 PCs and measured the step and tail current upon voltage changes of the membrane. We measured the tail current, due to blocking the vast majority of the non-HCN channels the tail current becomes a measure of the HCN-current deactivation, which reflects the amount of activated HCN channels in the membrane (Figure 7A). The block of the vast majority of all channels, except for the HCN channels, allows us to determine the HCN-current levels when the channels are closing. Upon using zatebradine, a specific HCN-channel blocker, all tail current disappeared, suggesting that we indeed only measured HCN-currents (Figure 7 and Appendix A). To determine the contribution of HCN1 and compare that to all HCNs, we used the HCN1 blocker propofol, which does not affect HCN2, HCN3, and HCN4 functioning. First, we determined the input, membrane, and series resistances of the patched cells (Figure 7 and Appendix A) after blocking all channels, except HCN. Tail current analysis revealed that the total HCN current in LIII PCs is significantly higher compared to LX PCs from −70 Vm and lower (e.g., at −70 mV: *p* = 0.0220, Figure 7(B1)). When we block the HCN1 current using the HCN1-specific blocker propofol, a clear decrease in HCN currents can be observed, reducing the difference between LIII and LX (Figure 7(B2)). Subtracting the tail currents in the presence of propofol from the total HCN currents (Figure 7(B3)), confirms that the HCN1 current is larger in LIII PCs at most membrane voltages from −80 Vm and lower. Since HCN1 currents can be detected at physiological membrane voltages, we performed cell attached recordings to measure the intrinsic firing rate and CV2 (Figure 7(C1–C3)) in >P30 PCs. LIII PCs have a higher firing rate and higher CV2 (Figure 7(C2,C3)); ‘aCSF’ group, data previously published in Beekhof et al. (2021), included for comparison) [26]. Firing rates are no longer significantly different between LIII and LX in the presence of propofol (*p* = 0.0565, Figure 7(C2)), but the difference in CV2 can still be confirmed (*p* = 0.0011, Figure 7(C3)). However, adding propofol does significantly increase the CV2, selectively in LX PCs (*p* = 0.0007, Figure 7(C3)). It should be noted that the presence of propofol also increased the firing rate in both LIII and LX PCs (Figure 7(C2)).

In addition to blocking HCN1, propofol also has an agonistic effect on the GABA_B_ receptor, which—perhaps counterintuitively—has been previously shown to increase the firing rate in nigral DA neurons [71]. To determine to what extent the propofol effects can be linked to blocking HCN1, we next also included the GABA_B_ receptor antagonist CGP to counter the agonistic effect of Propofol on GABA_B_. Adding CGP confirmed that the effect on firing rate and CV2 are largely related to the effects of GABA_B_ activation by propofol (Figure 7(C2,C3)).

Next, we aimed to determine if the AP shape is affected by propofol. In the presence of propofol, the differences in AP threshold, half width, rise time, and decay time are not affected by the presence of propofol (Figure 7(D,F1–F3), compared to Figure 1). The same holds for most of the other parameters, including the input, membrane, and series resistance, as also the AP peak (Figure 7 and Appendix A).

All in all, our results indicate that the HCN current is larger in LIII PCs, a difference that depends for a large part on HCN1. HCN1-blocker propofol affects multiple PC parameters, including the intrinsic firing rate and CV2 of PCs, but these effects could largely be rescued by abolishing the effect of propofol on GABA_B_.

## 4. Discussion

Here we investigated several basic electrophysiological features of PCs in adults and developing mice in relation to their zonal location. We found that APs of LX are larger and slower, with higher amplitude, half-width, decay time, and duration of the AHP, and also have a higher membrane resistance. Analysis of juvenile mice (P15–17 and P22–24) only confirmed a higher decay time between PCs of the two lobes, while the AP shape of PCs at P3–P5 was dramatically different from all other ages, indicating the undeveloped status of these neurons. We showed that the integration of both hyperpolarizing and depolarizing currents is lobule dependent, which for hyperpolarizing currents can already be seen in P15–P17 PCs. Next, we injected negative and positive current ramps to investigate the history dependent current integration. We found that PCs in LIII and LX from mice at >P30 and P15–P17 exhibit this history dependent current integration. In fact, for both the firing rate and the membrane potential the previous state of the cell determines the effect of current injection. In search off the underlying mechanism, we tested the role of HCN. We found higher levels of HCN in general, and HCN1 in particular, in LIII PCs of >P30 mice. HCN1 indeed causes a higher intrinsic firing rate in PCs and increases the CV2 in LX, but we could not link it to the AP shape.

It has been shown that P3–P5 PCs are morphologically immature and have a low intrinsic firing rate [26,80], and that the capacity to fire sequential APs is critically improved after the eye opening of postnatal mice [57]. Here, we found that the kinetics of the AP in P3–P5 PCs are, compared to >P15 PCs, very slow, confirming that the PCs are also physiologically immature at P3–5. One big electrophysiological change during late embryonal/early postnatal development is the GABA switch, which is the moment that GABA switches from being excitatory to inhibitory. The timing of the switch depends on the brain area and neuronal type [84]. PCs of P3–P5 do not yet form a monolayer [85]. A sign for the GABA switch to have happened is that the neurons are fully migrated to their final location. Thus, we can assume that at P3–P5 PCs GABA is excitatory and at the other age groups we assumed that GABA is inhibitory [86,87]. This major event is an indicator for different ion concentrations inside and outside the cell, thus it is not surprising that the AP shape is clearly different from the other age groups we recorded. PCs in rat from P3 and P9 exhibit a slow sodium discharge, increasing conductance till P18, and show a matured sodium AP at P21 compared to P90. Together with the maturation of the rise time, the decay time kinetics are also increasing with the same age dependency, while at P18 a fast hyperpolarization is detected, and matured at P21. McKay and Turner (2005) similarly found a decrease in AP threshold and input resistance from P3 to adult, and an increase in AP amplitude and rheobase, while at P3 PCs lack a fAHP [80]. PC morphology is intricate and complex, prohibiting accurate assessment of the membrane capacitance using our voltage-clamp protocol [88,89,90,91]. Therefore, we decided not to calculate the Cm.

Here, by separately recording LIII and LX we were able to determine which properties start to differentiate in development and at what time. It should be noted though, that the relatively small number of LIII recordings at P3–P5 hampers the comparison of LIII and LX PCs at that age. However, given that we previously found that PCs in LIII have an intrinsically higher firing rate from P12 onwards compared to LX [26], we consider it unlike that there would be overt differences in parameters at the immature stage of P3–P5 and this is also not suggested by the data we obtained. Analysis of the AP kinetics supports the concept that, at least from P15 changes in AP kinetics facilitate this difference in firing rate. From P15 onwards, PCs in LIII have a faster decay time compared to LX PCs, while the rise time is the same between both lobules from P15. While we could not detect an action potential threshold and rise time difference, we observed a higher AP amplitude and AP peak in >P30 PCs in LIII versus LX. This could be because of two reasons: (1) the LX PCs have a faster discharge of sodium by expressing more sodium channels or sodium channels with faster conductance and/or (2) a smaller neuron size in LIII PCs causes a bigger voltage change upon the same amount of sodium discharge. Given that previous work indicated that LX PCs are smaller than LIII PCs [26], we consider the former option more probable, but this remains to be determined. Similarly, the larger rheobase in LIII PCs could be explained by a larger neuronal size, as it takes more current to elicit an AP.

The fAHP repolarizes and follows an action potential, and it is carried by large conductance Ca^2+^-activated K^+^ (BK) channel [92]. Our data show that the fAHP is not present in P3–P5 PCs, and appears in the P15–P17 and P22–P24 PCs. The level of the fAHP between LIII and LX is only different in >P30 cells, at which point LIII PCs have the highest fAHP amplitude. Interestingly, the kinetics of the fAHP differs between LIII and LX in >P30 and P22–P24 mice; whereas the amount of current is the same or bigger in LIII PCs, the time to reach the maximum amplitude is lower in LIII compared to LX. The BK current is important for fast AP firing in PCs [93], and not all BK currents are sensitive for the blocker Iberiotoxin. Consequently, PCs have two distinct BK currents with different conductance, a fast activating and slow inactivating Iberiotoxin-sensitive current and a slow activating non-inactivating Iberiotoxin-insensitive current [94,95]. The larger current amplitude that reaches the maximum amplitude faster, suggests that the Iberiotoxin-sensitive BK current is more dominant in LIII PCs.

The sAHP repolarizes and follows a burst, and can be subdivided into a medium (50–100 msec) and a slow component (1–2 s). The medium component of the sAHP is carried by SK channels and is influenced by BK channels [95], while the slow component of the sAHP is carried by calcium activated potassium voltage independent currents, but the channels carrying the slow component are unknown [75]. While at P15–P17 there was some evidence for a higher level of sAHP in LIII PCs, this was no longer observed by P30. Most striking is the larger sAHP current in the younger PC group, a developmental trajectory that has potential relevance as the sAHP plays a role mediating firing patterns [96]. The voltage sag is indicative for the presence of the Ih current [74], this current is not specific for a particular ion, and plays a small role for single action potential, but is important in the intrinsic firing cells [97]. Ih currents are also involved in the intrinsic excitability of PCs; less Ih current is associated with higher firing rates [98], and an increase causes a decrease in PC firing rate [99]. Our data indicate that Ih currents are less involved in the distinct firing rates between subpopulations, but are more important for the increase in the firing rate in PCs during development, by gradually decreasing the amount of Ih current.

Here we describe that intrinsic excitability, determined by calculating the number of action potential evoked by positive current steps, is higher in LIII PCs than in LX PCs. This finding is in line with previous work in which PCs from the same lobule but different Aldoc zones were found to be intrinsically more excitable in Aldoc− compared to Aldoc+ areas [59,100]. Interestingly, Viet et al. (2022) used mice from P16–P24, matching our excitability results from >P30 PCs, but contrasting with PCs from P15–P17 mice, which we found here not to differ. The absence of a difference appears to be primarily because LIII PCs from P15–P17 mice have a significantly lower firing rate compared to LIII PCs >P30, which is in accordance with our previous work showing that Aldoc− PCs reach their mature firing rate later than Aldoc+ PCs [26]. Taken together, we are able to more accurately pinpoint the time point when LIII PCs reach a higher intrinsic excitability than LX PCs to the end of the third postnatal week. When we normalize the firing rates in negative and positive current steps, the initial response in LX PCs is a faster decrease and increase in firing rate than those in LIII, which is in line with the higher input resistance in LX PCs. Interestingly, when we then analyzed the underlying changes in membrane potential, both for positive and negative current steps, we found only subtle differences. The difference in intrinsic excitability in response to positive steps only for larger steps correlated with a larger depolarization in LIII PCs, while interestingly LX PCs were hyperpolarized more in responses to negative current steps. These findings are opposite to the theory of Aldoc+ PCs being ‘upbound’ and Aldoc− PCs ‘downbound’ [48], suggesting that the explanation for responses detailed in that theory depend more on synaptic mechanisms, rather than intrinsic features. Moreover, we find that the differences in firing frequency and membrane potential do not occur at the same amount of injected current, suggesting that there is no one-to-one causal link between the membrane potential of the neurons and the expressed firing rate. Thus, with only the firing rate we cannot predict the membrane potential, and vice versa. This is in line with previous work [76] and indicates a more complex relationship between membrane potential and firing rate, which is addressed further below.

Finally, we aimed to elucidate if the previous state of a PC influences the membrane potential or firing rate resulting from a specific level of current input. Two previous studies reported the presence of this history-dependent current integration in PCs of P18–P25 mice [58,101], while another found in P21–P56 PCs that current integration is history-independent, unless the HCN1 channel is genetically deleted from PCs [76]. Our results match the results of Williams et al. (2002) and Buchin et al. (2016) [58,101], in that the firing frequency and membrane potential are dependent on the previous state of the PC. Moreover, although the results in this manuscript show that P15–P17 PCs are in multiple fundamental ways different from >P30 PCs, they already have developed most features of adult history dependent integration of currents. This suggests that history-dependent integration is a fundamentally relevant feature of PCs. The work on HCN1 of Nolan et al. (2003) suggests a prominent role for HCN in history-dependent current integration [76].

HCN1 is considered to have a prominent role in learning and incurrent integration in cerebellar PCs [76,83,102]. The typical current reversal potential of a HCN channel is approximately −30 mV, which is above the AP threshold of PCs. Therefore, HCN currents depolarize and could contribute to maintaining high frequency firing [103]. HCN function, however, is complex, as blocking and application of neuromodulators can decrease or increase the conduction velocity, and it is unknown which subtype of HCN is responsible for these effects [104]. Moreover, it is unknown if HCN involvement is lobule specific. Therefore, we measured the total amount of HCN current, HCN1-4, and the HCN1 current in ‘adult’ (<P30) PCs in LIII and LX. We found that LIII PCs exhibit a larger HCN current than LX PCs, suggesting indeed that the HCN current contributes to the higher firing rate of LIII PCs. Using the only subtype-specific HCN blocker propofol, a HCN1 blocker, allowed us to differentiate between HCN1 and HCN2-4 currents. Our data suggest that both the HCN1 as well as the HCN2-4 component are larger in LIII PCs than in LX PCs. Surprisingly, blocking HCN1 channels increases the firing rate of both LIII and LX PCs. Previous work found that blocking all HCN channels results in a decrease in firing rate [104], which could imply that HCN1 has a different role in neurons than the other HCN subtypes. Additional experiments are necessary to unveil the role of different subtypes of HCN in PCs and to uncover the functioning in the developmental pathway.

Taken together, this study highlights that PCs in different lobules are intrinsically different in fundamental parameters, and that these differences often originate in development. We opted to test LIII and LX PCs because these populations virtually completely match Aldoc− or Z− and Aldoc+ or Z+ PCs, respectively. It should be noted though that this approach cannot exclude the possibility of further refinement of parameters in other lobules or that there is a lobule-specific element to the differentiation. Nonetheless, several recent studies have found robust and reproducible differences between Aldoc− and Aldoc+ populations, supporting the relevance of this differentiation in understanding cerebellar function [18,28,29,105]. Here we find that key cell physiological features such as AP shape, negative and positive current integration, and HCN currents differ between the two subpopulations. In contrast, history dependent current integration is not zone- or age-dependent in 15 out of 16 comparisons (except for the effect of positive currents on membrane potential in juvenile PCs of LIII), suggesting that this is a crucial parameter in the cerebellar circuit. These results imply that certain features, including types of input and history-dependent current integration, are generic, while others, including AP shape, firing rate, specific currents, but also different types of plasticity [59,106], are sub-population dependent. Uncovering the functional relevance of differences in subpopulations of PCs, both in the adult as well as the developing cerebellum, is crucial for understanding cerebellar function. Knowledge of the functional mechanisms will be essential to gain insight in cerebellum-related disorders, which are commonly related to development. Future experiments should elucidate the developmental timeline of all fundamental parameters in order to reveal how cerebellar development supports the initiation of proper, or disrupted, sensorimotor integration.

## Figures and Tables

**Figure 1 cells-12-00623-f001:**
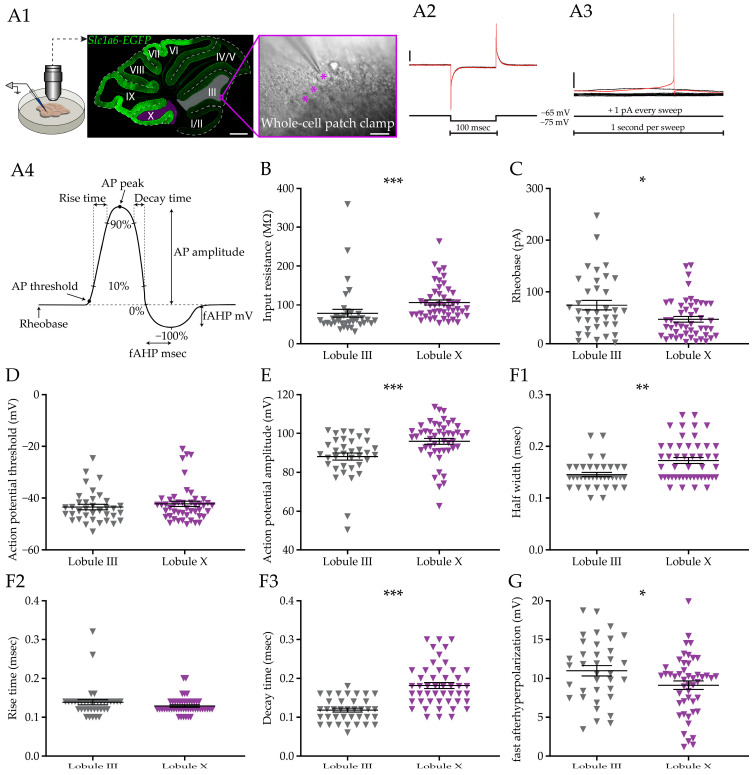
Cellular properties and Purkinje cell action potential shape in different subpopulations. (**A**), Schematic depiction of the recording approach with example image of recorded neurons and a view of a sagittal cerebellar brain slice of a *Slc1a6-EGFP* mouse (Roman numerals indicate lobule number) (**A1**) and methods to determine various parameters, including negative currents pulses (**A2**), slowly ramping holding current (**A3**) and demarcation of various parameters (**A4**). (**B**–**E**), Analysis of input resistance, rheobase, action potential threshold, and AP amplitude for each subpopulation in mice of >P30. (**F**), Action potential duration analysis, examining half width (**F1**), rise time (*p* = 0.1621) (**F2**) and decay time (**F3**) for each subpopulation. Note that, due to their low values, these parameters reveal the sample rate (50 kHz), resulting in a step-wise patterns of values. (**G**), Plot of the amplitude of the fast after hyperpolarization (in mV). Scale bar in A1-*Slc1a6-EGFP*, 100 μm; (**A1**) Whole-cell patch clamp 30 μm; (**A2**) vertical scale bar 200 pA; (**A3**) vertical scale bar 20 mV. Asterisks denote: * for *p* < 0.05, ** for *p* < 0.01 and *** for *p* < 0.001. See Appendix A for all statistical data.

**Figure 2 cells-12-00623-f002:**
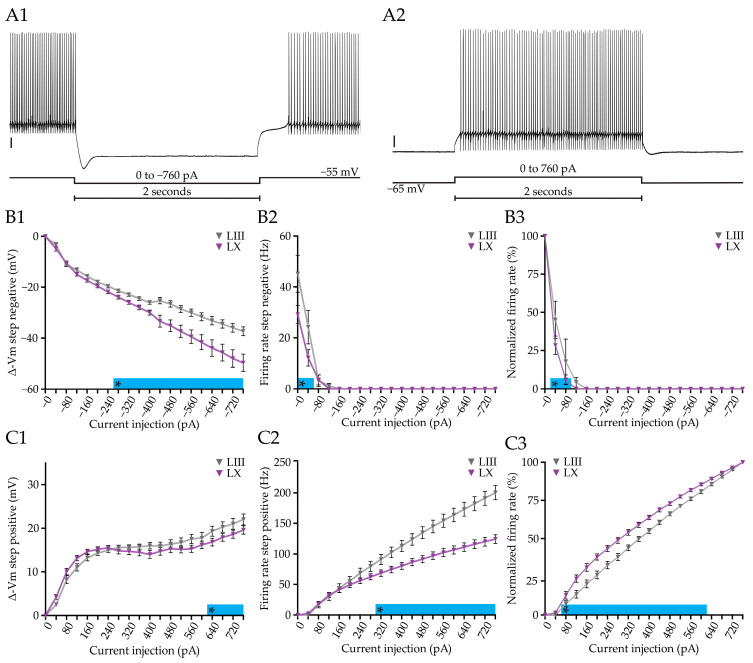
Positive and negative current injections in Purkinje cell from lobules III and X. (**A**), example traces of negative (**A1**) and positive (**A2**) current steps, initiated from a resting membrane potential of −65 mV and −55 mV, respectively. (**B**), Plots of the voltage change (**B1**) absolute firing rate (**B2**), and normalized firing rate ((**B3**), to the rate at 0 pA), induced by a range of negative current injections (0 to −760 pA). (**C**), Similar analysis of the effect of different positive current injections, starting from −65 mV, on membrane potential ((**C1**), plotted as voltage change), absolute firing rate (**C2**), and normalized firing rate ((**C3**), normalized to the value of +760 pA). Vertical scale bar in (**A1**) 10 mV; (**A2**) 10 mV. Asterisk denotes *p* < 0.05. See Appendix A for all statistical data.

**Figure 3 cells-12-00623-f003:**
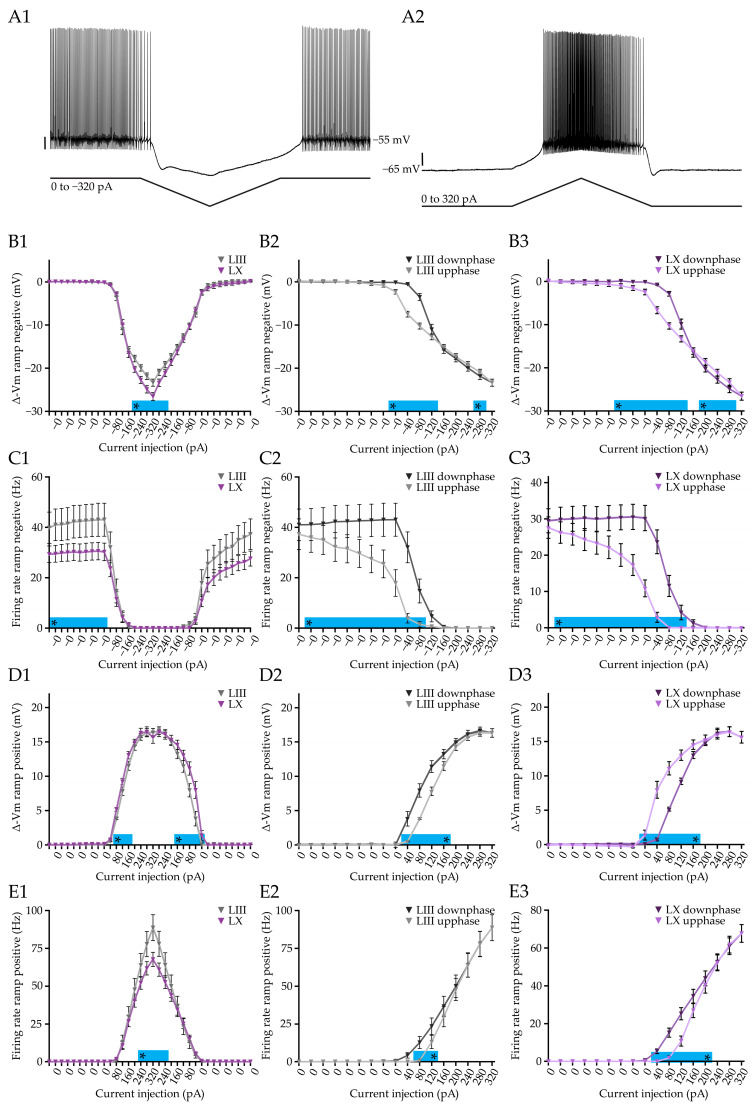
History-dependent integration of current in Purkinje cells. (**A**), Example traces for negative (**A1**) and positive (**A2**) current ramps, used to test history-dependent current integration. (**B**), Comparison of negative current ramps starting from a resting membrane potential of −55 mV between subpopulations (**B1**) and comparing the downphase and the upphase for LIII (**B2**) and LX (**B3**) PCs. (**C1**), Similar to (**B1**), but now for the firing rate. In (**B2**,**B3**,**C2**,**C3**) significant differences indicate that the voltage response and firing, respectively, are dependent on the previous state, and thus history-dependent. (**D**), Plots of voltage change induced by positive current ramps starting from −65 mV, comparing LIII and LX (**D1**) and the upphase and downphase for LIII (**D2**) and LX (**D3**). (**E1**,**E2**,**E3**), Similar to (**D1**,**D2**,**D3**), but now for the firing rate of the recorded PCs. Vertical scale bar in (**A1**), 10 mV; (**A2**) 10 mV. Asterisk denotes *p* < 0.05. See Appendix A for all statistical data.

**Figure 4 cells-12-00623-f004:**
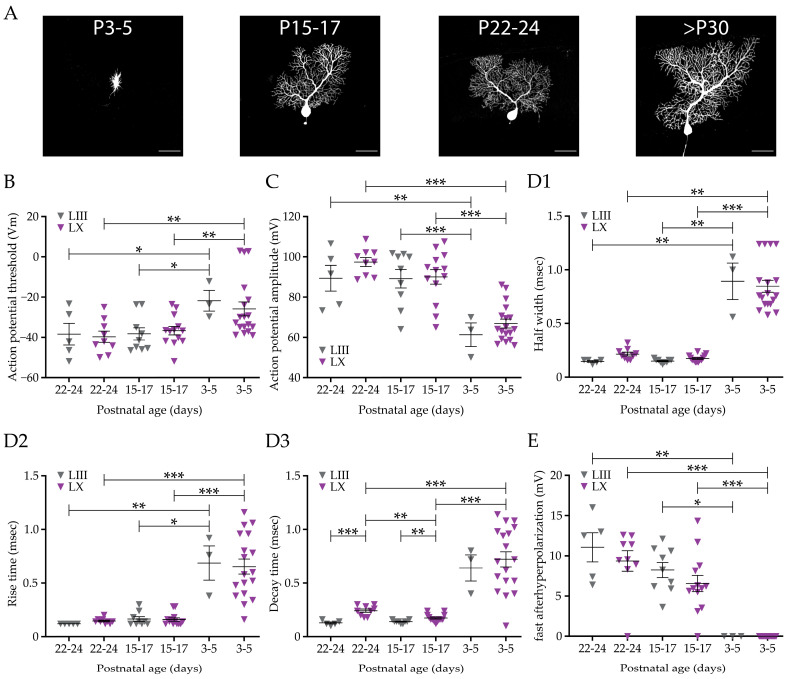
Development of cellular and action potential features compared between Purkinje cell subpopulations. (**A**), Fluorescence images of biocytin-filled PCs revealing the development of morphology from P3–5 until >P30. (**B**), Analysis of action potential threshold in P22–24, P15–17, and P3–5 days old mice, comparing PCs from LIII and LX. (**C**), Similar analysis for AP amplitude. (**D**), Developmental profile of action potential shape, comparing half width (**D1**), rise time (**D2**) and decay time (**D3**). (**E**), Analysis of the development of the fast after hyperpolarization for each subpopulation. See Figure 1 for comparison with >P30 mice. Scale bar in A1, 50 μm. Asterisks denote: * for *p* < 0.05, ** for *p* < 0.01 and *** for *p* < 0.001. See Appendix A for all statistical data.

**Figure 5 cells-12-00623-f005:**
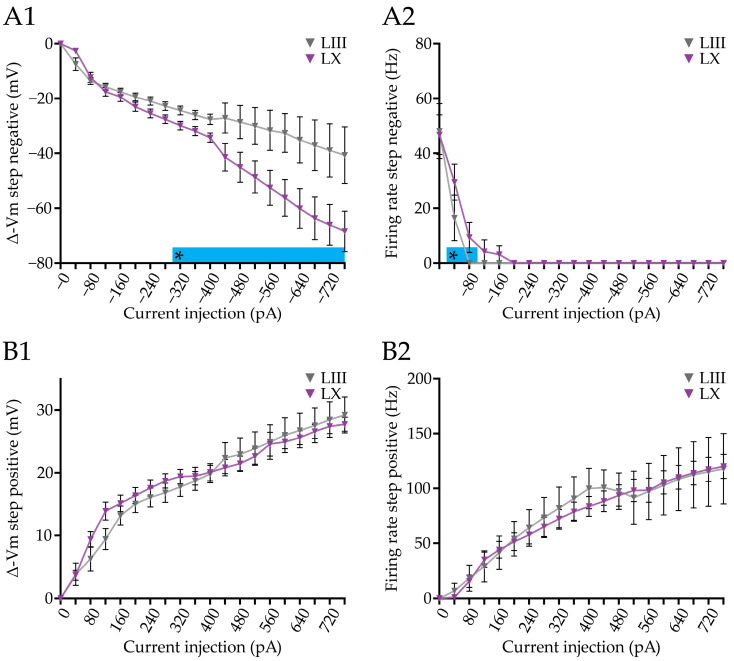
Responses to positive and negative current injections in Purkinje cells of juvenile mice. (**A**), Recordings from PCs in LIII and LX of P15–17 mice. Analysis of the change in membrane potential (**A1**) and firing rate (**A2**), induced by negative current injections (0 to −760 pA) starting from a membrane potential of −55 mV. (**B**), Similar analysis for the responses in change in membrane potential (**B1**) and firing rate (**B2**) to the injection of positive current pulses (0 to 760 pA) starting from a membrane potential of −65 mV. Asterisk denotes *p* < 0.05. See Appendix A for all statistical data.

**Figure 6 cells-12-00623-f006:**
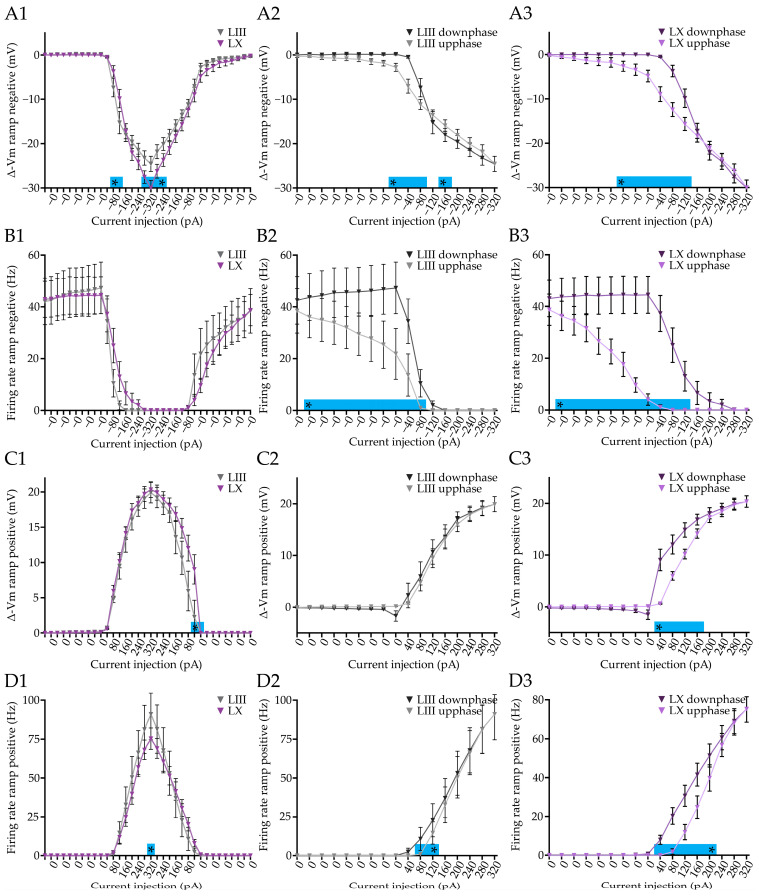
History-dependent current integration in Purkinje cells of juvenile mice. (**A**), Responses of subpopulations of PCs in juvenile mice (P15–17) to negative current ramps starting from a resting membrane potential of −55 mV (**A1**). Comparison of the downphase and the upphase for LIII (**A2**) and LX (**A3**) PCs. (**B**), Similar to (**A**), but now for the firing rate. In (**A2**,**A3**,**B2**,**B3**) significant differences indicate that the voltage response and firing, respectively, are dependent on the previous state, and thus history-dependent. (**C1**,**C2**,**C3**), Plots of voltage change induced by positive current ramps starting from −65 mV, comparing LIII and LX (**D1**) and the upphase and downphase for juvenile LIII (**D2**) and LX (**D3**) PCs. €, Similar to (**D**), but now for the firing rate of the recorded PCs. Asterisk denotes *p* < 0.05. Appendix A for all statistical data.

**Figure 7 cells-12-00623-f007:**
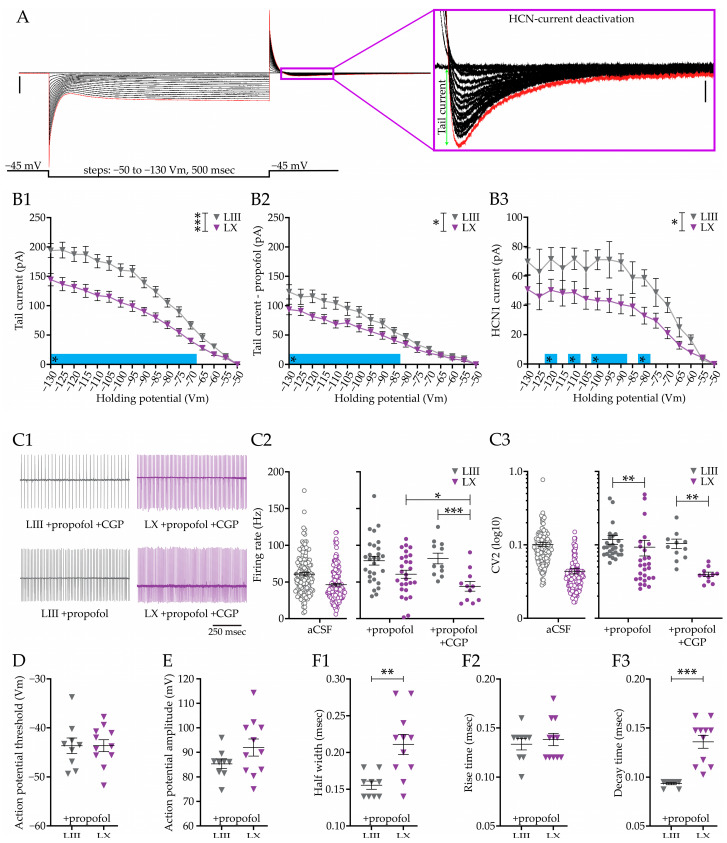
HCN currents in subpopulations of cerebellar Purkinje cells. (**A**), Analysis of HCN currents by holding PCs at −50 mV and imposing voltage steps, ranging from −55 to −130 mV, and calculating the amplitude of the release-evoked tail currents. (**B**), Tail currents analysis comparing the responses in PCs from LIII and LX of >P30 mice in normal aCSF (**B1**) and after adding HCN1 blocker propofol (**B2**). Subtraction of the B2 from B1 gives an indication of the amplitude of the HCN1 current (**B3**). (**C**), Effect of inhibiting HCN1 on PC firing properties using propofol and propofol + CGP, visualized by example traces (**C1**), analysis of the firing rate (**C2**) and the local coefficient of variance (CV_2_, (**C3**)). (**D**–**F**), Analysis of the effects of propofol on action potential threshold (**D**), AP amplitude (**E**), half width (**F1**), rise time (**F2**), and decay time (**F3**) in PCs from LIII and LX. See Figure 1 for comparison. Vertical scale bar in A 1000 pA; A-insert HCN-current deactivation 50 pA. Asterisks denote: * for *p* < 0.05, ** for *p* < 0.01 and *** for *p* < 0.001. Appendix A for all statistical data.

## Data Availability

The data presented in this study are available upon request to the corresponding authors.

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
