# Peer review of "Lobule-Related Action Potential Shape- and History-Dependent Current Integration in Purkinje Cells of Adult and Developing Mice"

_cells, 2023, doi:10.3390/cells12040623_

Round 1
Reviewer 1 Report
The authors conducted detailed electrophysiological analyses of Purkinje cells in lobule localized subpopulations from adult and developing mice. The authors chose lobule III (L III) and lobule X (L X) which are known to express Aldoc at low and high levels, respectively. They found differences in action potential shape and current integration between neurons in different lobules. Purkinje cells in both lobules demonstrated history dependent current integration. Many, but not all, lobule related differences were observed in developing Purkinje cells. There was a difference in HCN channel mediated currents between lobules, which may play a role in lobule specific electrophysiological properties. While previous studies have uncovered distinct firing and connectivity in lobule related subpopulations of Purkinje cells, precise analysis of physiological properties such as action potential shape has been limited, particularly in developing brains. This study provides a systematic and comprehensive description of distinct subpopulations of Purkinje cells, which will help our better understanding of a cerebellar neuronal circuit including the one during development.
Although this work will contribute to an advance in cerebellar research without doubt, I raise one major concern and provide several minor comments that the authors should address before publication.
Major point
The biggest concern is about the experiments recording HCN channels. The authors used previously published “control” data in Figure 7C2 and 7C3 to compare with the drug treated groups. While it is clearly mentioned in the text and the original article was properly cited, this is discouraged. While reanalyzing prior data in multiple papers may under some circumstances be ok, in this case where a data set from simple experiments and a paired comparison is made, it is against the standards of experimental design that are commonly held in the field to produce verifiable and reproducible results. Therefore, the comparison between aCSF group vs + propofol (+ CGP) groups in Figure 7C2 and 7C3 (Line 540 – 553 in the text and Line 589 – 591, 653 – 656, 705 – 713 in the discussion) is not valid. To do so, the authors should obtain a new set of data for aCSF group by conducting the same experiment interleaving control and drug recordings. My suggestion is that the authors either provide this new data set or simply remove this experiment as it does not make a critical change to the conclusions of the study.
Analyzing tail currents after voltage step protocol uncovered a lobular difference in HCN channel mediated current (Figure 7B). The authors further dissected HCN1 channel mediated current pharmacologically, and again found a lobular difference (Figure 7B). The authors reported lobular difference in firing rate in cell attached recordings (Beekhof et al. Ref [26]) and in whole cell recordings (Figure 2), both of which consistently demonstrated higher firing in L III than in L X. When HCN1 channels are pharmacologically blocked, lobular difference in firing cannot be detected in cell attached recordings (Figure C2: +propofol panel). While the effect of GCP makes the interpretation complicated, these data may be enough to draw the interpretation or conclusion that HCN1 channels play a role in electrophysiological parameters of Purkinje cells in different lobules.
Minor points
Line 38: 100.000 may be a typo for 100,000
Line 137: Please provide reference for Beekhof et al (2021)
Line 181: -760 pA may be a typo for +760 pA
Line 230, 231, and 233: It is not very clear what 1600 lowest measurements means. Does “1600” have any unit? Does it mean the lowest value among 1600 data points after the voltage step? Please clarify.
Line 278 – 280 and Line 406: Membrane capacitance (Cm) is also commonly used as a proxy for cellular size. As the authors measured input resistance using step pulse protocol in which Cm can be measured simultaneously, it may be helpful to present Cm as well.
Line 284: This may be an overstatement. Other possibilities such as Na channel expression level cannot be excluded.
Line 286 – 287: It may be better to use consistent terminology with the figure and the legend. “spike amplitude” and “spike maximum” may be better to be replaced with “AP amplitude” and “AP peak”.
Line 287 – 291: In Figure 1F1 – F3, data points shown as scatter plots look like distributing step-wise (lots of data points have same values while each data point in 1B – 1E has different values). It is not clear how this happens.
Line 329: Scale bars should be presented in Figure 2SA1 and 2SA2.
Line 382 – 383: Did the authors apply any statistical test to examine if this feature (history dependent integration) is different between subpopulations? If not, it might be better to simply say “this feature is present in both subpopulations” or so.
Line 406: In Figure 4 and 4S, it might be safer to have more cells (particularly L III cells of P3-P5) as the authors claim there is no difference in several parameters. Spike amplitude (4C) and spike maximum (4SB) might look different between L III and L X at P3-P5, for instance.
Line 527: “Blocking all channels, except for the HCN channels ~” might be an overstatement. Are all non-voltage-gated channels such as Ca2+ activated K+ channels also blocked? It might be safer to say “Blocking majority of voltage-gated channels” or so.
Line 552 – 553: It may be better to clearly say "blocking GABAB activation decreased firing rate" or so. Is there any explanation why blocking GABAB decreases, not increases, firing rate?
Line 569 – 571: In Figure 7C1, it is said “P6-9” and “P18-27”, but the legend says “Propofol” and “Propofol + CGP”.
Line 643: “were” is maybe a typo for “while” or “where” or so?
Line 725 – 726: During development, history dependent integration can be only partially observed (Figure 6). Particularly in positive current protocol it is absent in LIII (Figure 6C2). This statement needs to be revised.
Reviewer 2 Report
This manuscript from Beekhof and Schonewille tracks action potential waveform and intrinsic excitability in anterior and posterior cerebellum at different developmental timepoints. Their data illustrates innate differences in the baseline electrophysiologic characteristics of Purkinje neurons across lobules and identifies potential critical periods in early neuronal development that could be further explored in future studies. While many of these baseline characteristics have been illustrated in other studies to varying extents, the current study takes a focused and more comprehensive look at waveform and excitability across cerebellar regions. Discussions of history-dependent current integration and developmental critical periods are of particular interest.
I would make the following comments and suggestions:
1. The visual organization of this manuscript is very clear. I thank the authors for their efforts to make large, dense figures easy to read. Figure 3 is a good example of this.
2. The authors used a GFP reporter line that tracks with aldolase C expression for these studies. However, in the methods and results I only see that Purkinje neurons were visually identified for recording in a standard manner. Was GFP signal or absence verified before recording of each neuron? While lobule III largely contains zebrin negative cells, and lobule X is mostly zebrin positive, this is not true of every single neuron in each lobule and would be better to verify. Perhaps I missed this in the text, in which case it should be made more clear.
3. Similarly, did you record from any zebrin positive neurons in lobule III, or any zebrin negative neurons in lobule X? If so, how do the properties of these neurons compare to the overall population of zebrin positive or negative neurons in this study? In other words, are these firing differences entirely tied to zebrin expression, or simply to the lobule in which they are found?
4. I have some concerns about sample size for the data presented in Figure 4, particularly in the Lobule III P3-5 group and the Lobule III P22-24 group. I am not sure whether there is enough data to make proper comparisons across groups, and I would suggest adding more recordings from these two groups since this is one of the more novel areas of investigation in the present study. I agree with the overall interpretation that the action potential waveform appears very different in P3-5, but more recordings might help uncover spike waveform/excitability differences that are already present at P3-5 or if these develop later. This could also be acknowledged in the discussion.
5. I am slightly confused by the labels in Figure 7C1. Are these representative traces P6-9 neurons and P18-27 neurons? Or are these the same neurons before and after propofol +/- CGP? The text and figure labels are telling separate stories.
6. Lines 552-553: GABAb receptor antagonist referred to as “CPG” and later “CGP”, should be corrected.
Reviewer 3 Report
The Beekhof and Schonewille manuscript is examining the basal cell physiological features of two Purkinje cells (PC) in two locations (LIII and LX), and further investigate how these features are affected by development. Using in vitro electrophysiology, the authors showed that PCs in the LIII and LX displayed different electrophysiological properties, and these differences were present in the early developmental stage. The experiment is well designed and most of the data is properly interpreted. But the manuscript contains many small mistakes. Below are my major and minor concerns about this manuscript:
Below are my major and minor concerns about this manuscript:
Major concern:
-
The conclusion of Figure 2 is not convincing:
-
“ even when we equalize the membrane potential, firing rates are higher in LIII PCs, suggesting that membrane potential does not directly determine the firing frequency in the PCs of LIII and LX.” I don’t think the authors can get this conclusion from the data. The membrane potential of both PCs is the same only when giving 0 pA current injection. For non-zero current injections, because the input resistance of both PCs are not the same (Figure 1B), their membrane potential during current injection should be different.
-
“We also show that is prone to have larger changes in membrane potentials compared to LX upon the same current injection steps” Based on Figure 1B, the input resistance of LX PCs is larger than LIII PCs. According to the equation: V = I * R, the sma current injection should induce larger membrane changes of LX PCs. How could the authors explain this?
-
Considering the base firing of PCs from LIII and LX is different, it would be better to compare the relative frequency change for Figure B2 and C2.
Minor concern:
-
Line 8, No bold “P” for Purkinje cells.
-
Line 20, “that this” ?
-
Line 43, should give the abbreviation “CN” here, not in line 50.
-
Line 49, “,” should be “.”.
-
Line 68, “increase the capacity to spike and by lowering”.
-
Line 71, “Prukinje cells” to “PCs”.
-
Figure 1A1: please keep the name consistent. For the mutant mice, SLC1A6 is used, but here use the protein “EAAT4”.
-
Figure 1A1, the picture on the right should have a scale bar. In the legend, it said cells are filled with biocytin, but no biocytin labeling.
-
Figure 1A2 and A3 should have scale bars for the amplitude.
-
Figure 1A4 is great. But to show the difference between two groups, it would be great to plot APs from two groups together to show the difference.
-
Figure 1B, the unit of input resistance seems wrong.
-
Line 146, please explain the equation.
-
Line 260, should be “lobule III (LIII)”.
-
Line 263, “eGFP” or “EGFP”?
-
Line 277, what the “=<” mean.
-
Line 285, two “the shape of”.
Round 2
Reviewer 1 Report
The authors addressed all issues. Presenting previously published data in a separate panel as a reference, without direct statistical comparison with newly presented data, doesn't raise any concern.
Reviewer 2 Report
I thank the authors for addressing reviewer comments in the revised version of this manuscript. I am happy with the revisions and additions that have been made, and I believe they add clarity to the interpretation of the studies presented by the authors. I have no suggestions for further revision at this time.